# Effects of a Multicomponent Exercise Program, a Detraining Period and Dietary Intake Prediction of Body Composition of Frail and Pre-Frail Older Adults from the EXERNET Elder 3.0 Study

Ana Moradell [1,2,3,4], David Navarrete-Villanueva [1,2,3,5], Ángel Iván Fernández-García [1,2,3,4], Lucía Sagarra-Romero [6,7], Jorge Marín-Puyalto [1,2,3], Jorge Pérez-Gómez [8], Eva Gesteiro [3,9], Ignacio Ara [10,11], Jose Antonio Casajus [1,2,3,5,12], Alba Gómez-Cabello [1,2,3,4,12,13] and Germán Vicente-Rodríguez [1,2,3,4,12,*]

1   GENUD (Growth, Exercise, NUtrition and Development) Research Group, Universidad de Zaragoza, 50009 Zaragoza, Spain; amoradell@unizar.es (A.M.); dnavarrete@unizar.es (D.N.-V.); angelivanfg@unizar.es (Á.I.F.-G.); jmarinp@unizar.es (J.M.-P.); joseant@unizar.es (J.A.C.); agomez@unizar.es (A.G.-C.)
2   Instituto Agroalimentario de Aragón, CITA-Universidad de Zaragoza, 50059 Zaragoza, Spain
3   Red Española de Investigación en Ejercicio Físico y Salud en Poblaciones Especiales (EXERNET), 50009 Zaragoza, Spain; eva.gesteiro@upm.es
4   Department of Physiatry and Nursing, Faculty of Health and Sport Science (FCSD), University of Zaragoza, Ronda Misericordia 5, 22001 Huesca, Spain
5   Department of Physiatry and Nursing, Faculty of Health, University of Zaragoza, 50009 Zaragoza, Spain
6   Faculty of Health Science, Universidad San Jorge, 50830 Zaragoza, Spain; lsagarra@usj.es
7   Hospital General San Jorge, 2204 Huesca, Spain
8   HEME (Health, Economy, Motricity and Education) Research Group, Faculty of Sport Science, University of Extremadura, 10003 Cáceres, Spain; jorgepg100@unex.com
9   ImFine Research Group, Universidad Politécnica de Madrid, 28040 Madrid, Spain
10   GENUD Toledo Research Group, Universidad de Castilla-La Mancha, 13001 Ciudad Real, Spain; ignacio.ara@uclm.es
11   Biomedical Research Networking Center on Frailty and Healthy Aging (CIBERFES), 28029 Madrid, Spain
12   Centro de Investigación Biomédica en Red de Fisiopatología de la Obesidad y Nutrición (CIBERObn), 28029 Madrid, Spain
13   Centro Universitario de la Defensa, 50090 Zaragoza, Spain
*   Correspondence: gervicen@unizar.es; Tel.: +34-876-55-37-56

**Abstract:** The aging of humans is associated with body composition and function deterioration creating a burden on an individual level, but also on a societal one, resulting in an economic burden that is socially unsustainable. This study aimed to evaluate changes in body composition after a 6-month MCT (multicomponent training) and a 4-month detraining period, and to examine the possible influence of energy and macronutrient intake in these changes in frail and pre-frail older adults. A total of 43 participants from the training group (TRAIN) and 28 controls (CON) completed the study protocol. Body weight, body mass index (BMI), waist and hip circumferences, fat mass, fat free mass and fat mass percentage were recorded, with a bio-electrical impedance analyzer, at baseline, after 6 months and four months after finishing the MCT. A food frequency questionnaire was used to estimate energy intake. Mixed effect models did not show differences between groups. CON showed increases in hip circumference and waist (3.20 ± 1.41 and 3.06 ± 1.66 cm, respectively) during the first 6 months. TRAIN showed decreases in BMI (−0.29 ± 0.14), fat mass (−0.86 ± 0.38 kg), body fat percentage (−0.98 ± 0.36%) and increases in waist circumference (3.20 ± 1.41). After detraining, TRAIN group showed increases in fat mas (1.07 ± 0.30 kg), body fat percentage (1.43 ± 0.31%) and waist (3.92 ± 1.38 cm), and decreases in fat free mass (−0.90 ± 0.30 kg). CON group only showed an increase

in body fat (1.32 ± 0.47%). Energy intake was negatively associated with hip circumference in the first six months and fat mass during detraining in CON. Energy intake showed positive associations with fat mass in TRAIN during detraining. Only carbohydrates were negatively related to detraining changes in fat free mass and BMI in CON. In conclusion, the MCT reduces adiposity of frail and pre-frail older people, leading to a maintenance of fat free mass. In addition, these interventions should not be stopped in this population in order to improve health sustainability.

**Keywords:** frailty; adiposity; exercise; obesity; sarcopenia; energy intake

---

## 1. Introduction

The aging is associated with disease chronification and function deterioration placing a burden on an individual level, but also on a societal one, resulting in an economic burden that may be socially unsustainable. The ageing process is intrinsically associated with a wide variety of changes, including those related to body composition. Different factors work together leading to an increased fat mass (FM), decreased muscle mass and changes in body weight among seniors [1]. These changes are associated with an increased risk of suffering several pathologies and health problems, such as obesity, sarcopenia, sarcopenic obesity [1], metabolic syndrome [2], or even frailty. As those changes are related to a higher risk of hospitalization and adverse health outcomes, finding strategies to prevent them would drastically reduce healthcare costs [3].

Frailty in older people is a common clinical syndrome that involves an increased risk of poor health outcomes including falls, incident disability, hospitalization, and mortality [4]. Pre-frail identifies a subset at high risk of progressing to frailty and a potentially reversible condition before onset of established frailty [4]. Based on the current literature, observational studies have shown that body composition may have an important role in the risk and development of frailty in older adults. Specifically, Reinders et al. found that obesity and high waist circumference showed convincing results pointing towards an association with frailty [5]. Moreover, the phenotypic profile of frail older people is characterized by low muscle mass and a sarcopenic situation [5,6].

Specific exercise training programs, as well as physical activity, have been reported as one of the best non-pharmacological ways to improve health-related factors throughout life, including body composition parameters [7]. However, different types of training may lead to distinct health and body composition benefits, and not all populations may respond in the same way [8,9]. Nowadays, multicomponent training (MCT) programs are one of the most common interventions to improve health and body composition carried out in older people [10]. In this regard, a systematic review demonstrated that MCT positively impacts cardiorespiratory fitness, serum lipid profile, body composition, functional abilities, contributes to reduce the risk of falling and consequent hospitalizations [10]. Nevertheless, the effects of MCT on body composition of frail older adults have been investigated to a lesser extent and results are still controversial [11]. This fact may be partially explained by the wide differences in the methodologies followed among studies or by the mediation of important variables, such as those related with nutritional aspects. In this regard, energy imbalance and macronutrient distribution [12] induces changes in body composition [13] and, therefore, nutrition should be a factor to be considered in these studies in order to elucidate the real effects of this type of intervention on the body composition of frail older adults.

While many studies have focused on the effects of exercise in older people, it is still unclear if the interruption of training can reverse the health benefits obtained during the training program. Esain et al. reported an important decline in quality of life of older people following a 3-month detraining period [14]. In relation to body composition, there is no scientific evidence about the effects of a detraining period following a training program in this special population. Based on this scenario,

it may be interesting to design useful and plausible strategies to prevent or at least attenuate the return to pre-training conditions among elders.

Therefore, the objectives of this study were: (1) to analyze the effects of a 6-month MCT and 4 months of detraining on the body composition of frail and pre-frail older adults, within and between groups, and (2) to examine if nutritional intake is related to the changes observed after the training and detraining periods.

## 2. Materials and Methods

### 2.1. Study Design and Participants

This experimental study was carried out between 2018 and 2020 on the framework of the EXERNET-Elder 3.0 project. The complete methodology of the intervention is described in detail by Fernandez-Garcia AI, et al. [15]. Participants were recruited from three healthcare centers and three nursing homes from the city of Zaragoza, Spain. The inclusion criteria were people above 65 years categorized as frail or pre-frail according to the cut-off points of the Short Physical Performance Battery (SPPB) [16]. Those who had cancer and/or dementia were excluded. In total, 169 elders were initially derived from the centers mentioned above, and finally 110 met the inclusion criteria and were included in the sample as showed in Figure 1.

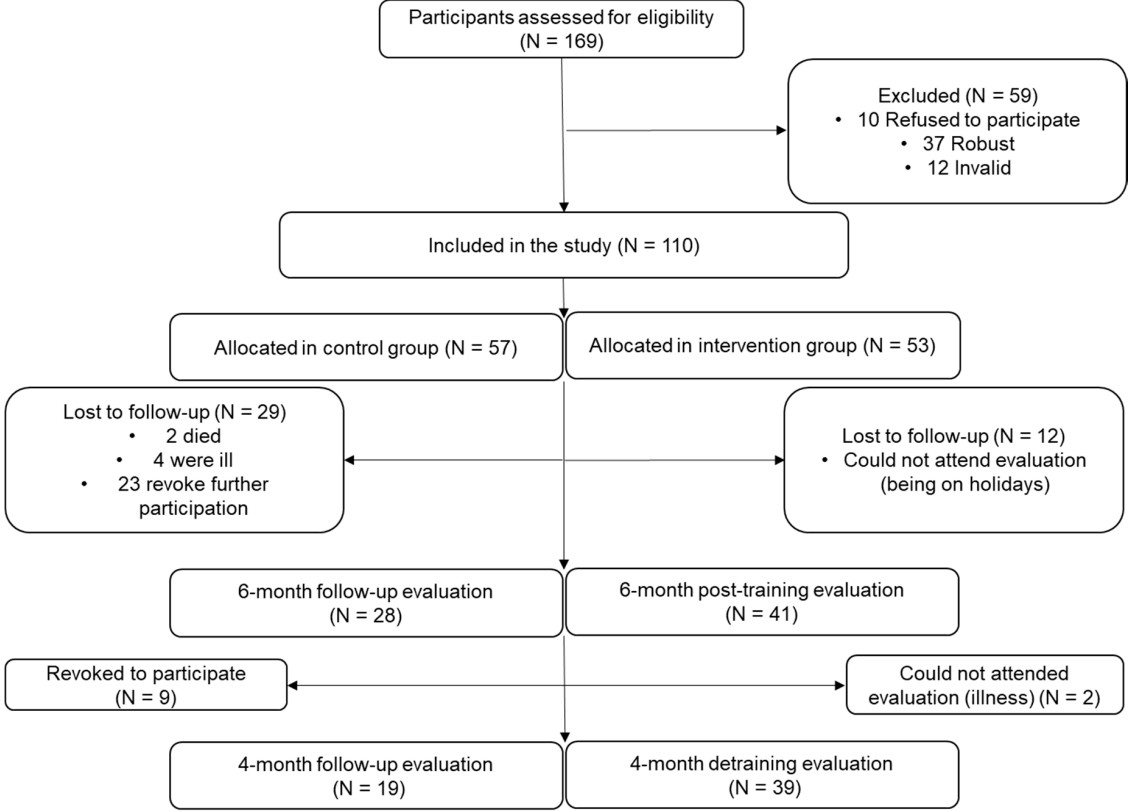

**Figure 1.** Flow chart of participant recruitment and follow-up.

Participants were then allocated by convenience into two groups, taking into account their preferences and availability: the control group (CON: $n = 57$; 43 females) and the training group (TRAIN: $n = 53$; 35 females). Both groups were followed at three particular occasions: after completing half MCT (3 months after the beginning), after the 6-month period lasting the MCT and 4 months later, for the evaluation of the detraining period. In this report, we have included data from the initial, the 6-month evaluation and the detraining, in order to study the final effect of MCT. Finally, 58 participants (39 TRAIN) completed the study protocol (Figure 1).

Participants of both groups (CON and TRAIN) received three talks related to healthy habits in order to improve the adherence to this study, especially in CON, during the 6-month training period. The topics of these talks were the following: "Functional capacity and frailty", "Nutritional recommendation for older people", "Physical exercise to improve health in older people". Moreover, in order to encourage participation among the TRAIN group, those who attended to the highest percentage of sessions received sport equipment as an incentive.

Personal information and other health outcomes were collected through a structured questionnaire, the complete set of studied variables is available elsewhere [15]. Specifically, the variables included in this report were: mean of daily walking hours, mean of sitting daily hours [17], smoking habits, Instrumental Activities of Daily Living Scale [18], Barthel Index [19] and Mini Nutritional Assessment [20]. Afterwards, researchers performed body composition measurements and physical fitness assessments. All of these measurements were performed three times at pre-training, 6 months later at the post-training and 4 months later at the detraining. Dietary record was obtained once at the middle of the intervention timeline.

All potential participants received oral and written information about the aims, possible benefits and risks derived from participation in this study. Afterwards, written informed consent was obtained from all the included participants.

The study was performed in accordance with the Helsinki Declaration of 1961 revised in Fortaleza (2013) [21] and the current legislation of human clinical research of Spain (Law 14/2007). The study protocol was approved by the Hospital Universitario Fundación de Alcorcón (16/50). This study was registered in the electronic repository clinicaltrials.gov (reference number: NCT03831841).

### 2.2. The EXERNET Elder 3.0 Multicomponent Exercise Program

Detailed information about the training protocol is described elsewhere [15]. In brief, the training protocol carried out in this study consisted of an MCT of 6 months in duration. Sessions were performed on three days a week for one hour, combining exercises in order to improve endurance, strength, flexibility, balance, coordination and specific daily life activities. All sessions included 10 min of warm-up, 35–40 min of main part exercises and 10–15 min of cool down, involving flexibility, breaths control and a cognitive exercise. During the whole intervention period, there was a progression of the training load to ensure that the stimulus was appropriate to cause adaptations in the body. Moreover, each session was adapted to different levels in order to individualize exercises depending on participant's characteristics and frailty status. Exercise groups were composed by 8–16 elders depending on the space available in the training center and all the sessions were supervised by specialized instructors complying a maximum ratio of 12 participants per instructor.

### 2.3. Short Physical Performance Battery (SPPB)

Physical performance and functional status were measured with the SPPB according with level of balance (ability to stand up for 10 s with feet positioned in three ways: together side-by-side, semi-tandem and tandem positions), usual gait speed (time to complete 4 m walk) and lower limb strength (time to rise 5 times from a chair). Each task was scored from 0 to 4, with 4 being the best, with a total battery score of 12 points (p) and four functional stages: dependent (<4 p), frail (4–6 p), pre-frail (7–9 p) and robust (>9 p) [16].

### 2.4. Anthropometric Measurements

Training workshops were organized to harmonize the assessment of anthropometric measurements before starting the study.

### 2.4.1. Height

A portable stadiometer with 2.10 m maximum capacity and 1 mm error margin (Seca, Hamburgo, Germany) was used to measure height. Subjects stood barefoot with their scapula, buttocks and heels resting against a wall; the neck was held in a natural non-stretched position; the heels were touching

each other with the toe tips spread to form a 45° angle; and the head was held straight with the inferior orbital border in the same horizontal plane as the external auditory tube (Frankfort's plane).

### 2.4.2. Waist and Hip Circumference

Waist circumference was taken at the level of the narrowest point between the lower costal border and the iliac crest. In case it was not appreciated, measure was taken between these points. Hip circumference was measured at the level of the greatest posterior protuberance of the buttocks. Both perimeters were taken using a flexible non-elastic measuring tape Rosscraft Anthrotape (Rosscraft Innovations Inc., Vancouver, BC, Canada). All anthropometrics were performed by International Society for the Advancement of Kinanthropometry (ISAK) accredited researchers following its standards [22]. Measures were performed twice, and the mean was calculated. If there were incongruences between the first two evaluations, a third measure was done, and the median was calculated.

### 2.5. Body Composition Measurements

A body composition analyzer based on Bio-Electrical Impedance Analysis (BIA) with 200 kg maximum capacity and 50 g error margin (TANITA BC-418MA, Tanita Corp., Tokyo, Japan) was used to measure the body weight (kg) and to estimate the body FM, the percentage of body fat (BF%) and the fat-free mass (FFM). Participants removed their shoes and heavy clothes before weighing. Body mass index (BMI) was calculated dividing weight (kg) by squared height ($m^2$). All measurements were performed at same hour and same conditions for all participants and evaluations.

### 2.6. Dietary Intake

A semiquantitative food-frequency questionnaire previously validated in Spain [23,24] was used to assess dietary intake. It was evaluated in the middle of the intervention timeline, just before the third month of intervention. The questionnaire included 137 items showing their typical portion size. Daily intake was calculated by multiplying the portion size by the frequency of consumption (nine options ranging from never/almost never to six times or more per day). Nutrient intake was estimated using Spanish food composition tables and other sources of information [25,26]. Variables considered in this study were the total consumption of kilocalories per day (Kcal/day) and carbohydrate, fat and protein percentages of the total intake (%).

### 2.7. Statistical Analysis

The Statistical Package for the Social Sciences v. 20.0 for Windows (SPSS, Inc., Chicago, IL, USA) was used to analyze the data. Normality of the sampling distribution was proved using the Shapiro–Wilk test. Statistical significance was set at level $p < 0.05$ in all tests. As age was not correlated with any changes found in body composition, it was not used as a covariate in any of the analyses.

Descriptive data are reported as mean and standard deviation (SD), number of participants (*n*) or percentage (%), according to the nature of each variable.

Mixed effect model analyses were performed to compare differences in changes during training, de-training and between pre-training and de-training, within and between groups. As no sex by time interaction was found for any variable, these analyses were carried out with men and women pooled together. Changes in body composition variables were calculated by subtracting post-intervention values minus baseline values. The same process was done to obtain changes between post-intervention and post-detraining, and pre-intervention-and post-detraining for all variables.

Afterwards, simple linear regression analyses were performed to estimate the contribution of daily intake (independent variables) on body composition changes (dependent variables) during training and detraining periods in both groups. Specifically, one linear regression analysis was performed with each independent variable for each dependent variable.

## 3. Results

Adherence to training averaged 83.2 ± 10.6%. There were no adverse effects and no health problems due to the MCT in the TRAIN group over the 6-month intervention period. The final sample included a total of 71 participants (28 CON (21 females) and 43 TRAIN (30 females)). Descriptive characteristics of the groups at baseline are shown in Table 1.

**Table 1.** Baseline descriptive characteristics of the sample.

| Characteristic | Control (*n* = 57) | Training (*n* = 53) |
|---|---|---|
| Age (year) | 80.4 ± 5.6 | 80.9 ± 6.1 |
| Sex | | |
| Male | 14 (24.6) | 18 (34.0) |
| Female | 43 (75.4) | 35 (66.0) |
| SPPB-Frailty status | | |
| Frail | 16 (28.1) | 14 (26.4) |
| Pre-frail | 41 (71.9) | 39 (73.6) |
| Walking hours per day | 1.13 ± 0.9 | 1.9 ± 1.4 |
| Sitting hours | 6.3± 2.3 | 6.2± 3.0 |
| Smoke | | |
| Yes | 4 (7.0) | 1 (1.9) |
| No | 51 (89.5) | 52 (98.1) |
| IADL groups | | |
| Moderate dependence | 10 (17.5) | 5 (9.4) |
| Mild dependence | 9 (15.8) | 14 (26.4) |
| Completely autonomy | 38 (66.6) | 34 (64.1) |
| Barthel Index | | |
| Moderate dependent | 1 (1.7) | 0 (0.0) |
| Mild dependent | 28 (49.1) | 20 (37.7) |
| Independent | 28 (49.1) | 33 (62.2) |
| MNA | | |
| Malnourished | 6 (11.3) | 2 (4.0) |
| At risk of malnutrition | 12 (22.6) | 14 (28.0) |
| Normal nutritional status | 35 (66.0) | 34 (68.0) |

Number of participants of the sample (*n*) and % per group for categorical variables; mean and standard deviation (S.D.) for continuous variables. SPPB: Short Physical Performance Battery; IADL: Lawton instrumental activities of daily living. MNA: Mini Nutritional Assessment.

*Effects of the Exercise Program and the Detraining Period on Body Composition*

Table 2 shows differences in changes between and within groups at the three evaluations.

**Table 2.** Body composition changes in control and training groups between pre-training, post-training and post-detraining periods.

| | 6 Months Training | | | 4 Months Detraining | | | Total 10 Months | | |
|---|---|---|---|---|---|---|---|---|---|
| | **CON** **(*n* = 28)** | **TRAIN** **(*n* = 43)** | ***p* Value** | **CON** **(*n* = 19)** | **TRAIN** **(*n* = 39)** | ***p* Value** | **CON** **(*n* = 17)** | **TRAIN** **(*n* = 43)** | ***p* Value** |
| Weight (kg) | −0.03 ± 0.40 | −0.57 ± 0.33 | 0.278 | 0.26 ± 0.35 | 0.14 ± 0.24 | 0.691 | 0.14 ± 0.61 | −0.61 ± 0.39 | 0.309 |
| BMI (kg/m$^2$) | −0.07 ± 0.18 | −0.29 ± 0.14 * | 0.246 | 0.19 ± 0.15 | 0.06 ± 0.10 | 0.570 | 0.01 ± 0.29 | −0.23 ± 0.18 | 0.496 |
| FM (kg) | −0.20 ± 0.42 | −0.86 ± 0.38 * | 0.180 | 0.39 ± 0.53 | 1.07 ± 0.30 * | 0.421 | 0.59 ± 0.53 | 0.11 ± 0.33 | 0.447 |
| FFM (kg) | 0.13 ± 0.38 | 0.43 ± 0.30 | 0.511 | −0.64 ± 0.39 | −0.90 ± 0.30 * | 0.618 | −0.40 ± 0.42 | −0.75 ± 0.26 * | 0.491 |
| BF% | −0.18 ± 0.45 | −0.98 ± 0.36 * | 0.144 | 1.32 ± 0.47 * | 1.43 ± 0.31 * | 0.895 | 0.84 ± 0.56 | 0.50 ± 0.35 | 0.606 |
| Waist Cir (cm) | 3.20 ± 1.41 * | 3.04 ± 1.15 * | 0.787 | 0.51 ± 2.07 | 3.92 ± 1.38 * | 0.184 | 4.08 ± 0.95 | 5.08 ± 1.04 | 0.487 |
| Hip Cir (cm) | 3.06 ± 1.22 * | 1.66 ± 0.85 | 0.252 | −0.50 ± 0.81 | −1.06 ± 0.55 | 0.719 | 1.80 ± 1.38 | 1.26 ± 0.82 | 0.736 |

Mean differences and standard deviation reported for each body composition variable. Statistical significance was established at <0.05. BMI: body mass index, FM: fat mass, FFM: fat free mass, BF%: body fat mass percentage, Cir: circumference. * Statistical significance within groups over time. *p* values describe differences between groups.

CON group showed increase in time for hip circumference, from 103.0 to 107.0 cm and an increase in waist circumference from 92.5 to 95.8 cm during the first 6 months corresponding to the training period (both $p < 0.05$). Afterwards, during 4 months of detraining, this group showed a statistical increase in body fat mass percentage (BF%) (from 39.0 to 40.3% ($p > 0.05$)).

The TRAIN group showed statistical decreases for BMI (from 29.4 to 29.0), FM (from 28.8 to 27.8 kg), BF% (from 38.2 to 37.2%) (all $p < 0.05$). Moreover, this group also showed increases during the training period in waist circumference (from 94.4 to 97.1 cm; $p < 0.05$). The 4-month detraining period resulted in TRAIN group increases in FM (from 27.8 to 28.9 kg), BF% (from 37.2 to 38.7%) and in waist circumference (from 97.1 to 99.0 cm); as well as a decrease in FFM (from 45.9 to 45.0 kg; $p < 0.05$). When pre-training and post-detraining was compared, a decrease in TRAIN group for FFM was observed (from 45.6 to 45.0 kg). No other variables showed statistical significance when compared these last evaluations ($p > 0.05$).

No differences were found between groups neither at baseline nor in the post-training or post-detraining evaluations (all $p > 0.05$).

In CON, total energy was positively associated with hip circumference during the training period. In addition, total energy intake was negatively associated with changes observed in FM and BF% during detraining. Moreover, in this group, carbohydrates were also negatively associated with changes during detraining in BMI and FFM. In TRAIN, positive associations between total energy intake and FM and BF% were found. No other variables related to macronutrient distribution were related to the variation of any body composition parameters neither during the training period nor in the detraining for any of the groups, as shown in Table 3 (all $p > 0.05$).

**Table 3.** Energy and macronutrient intake body composition predictive values in control and training groups.

| | | CONTROL | | | | | | | | TRAINING | | | | | | | |
| --- | --- | --- | --- | --- | --- | --- | --- | --- | --- | --- | --- | --- | --- | --- | --- | --- | --- |
| | | Training (*n* = 28) | | | | Detraining (*n* = 19) | | | | Training (*n* = 41) | | | | Detraining (*n* = 39) | | | |
| | | Total Energy (kcal) | CH (%) | Fat (%) | Prot (%) | Total Energy (kcal) | CH (%) | Fat (%) | Prot (%) | Total Energy (kcal) | CH (%) | Fat (%) | Prot (%) | Total Energy (kcal) | CH (%) | Fat (%) | Prot (%) |
| **Weigh (kg)** | β standardized | −0.081 | 0.127 | −0.122 | 0.046 | −0.048 | −0.377 | 0.321 | 0.022 | −0.005 | −0.104 | 0.149 | −0.166 | 0.108 | 0.074 | −0.220 | 0.269 |
| | *p* value | 0.653 | 0.482 | 0.499 | 0.801 | 0.824 | 0.069 | 0.127 | 0.918 | 0.974 | 0.495 | 0.328 | 0.275 | 0.506 | 0.649 | 0.172 | 0.093 |
| **BMI (kg/m²)** | β standardized | −0.058 | 0.166 | −0.166 | 0.012 | −0.063 | −0.513 | 0.395 | 0.166 | 0.108 | −0.116 | 0.178 | −0.190 | 0.029 | 0.090 | −0.300 | 0.461 |
| | *p* value | 0.769 | 0.399 | 0.399 | 0.953 | 0.792 | 0.021 | 0.085 | 0.484 | 0.538 | 0.507 | 0.306 | 0.274 | 0.874 | 0.618 | 0.090 | 0.007 |
| **FM (kg)** | β standardized | 0.182 | 0.288 | 0.106 | −0.023 | −0.476 | 0.077 | −0.125 | 0.017 | −0.147 | −0.202 | −0.099 | −0.169 | 0.338 | −0.101 | 0.099 | −0.070 |
| | *p* value | 0.327 | 0.116 | 0.571 | 0.904 | 0.019 | 0.719 | 0.561 | 0.938 | 0.347 | 0.194 | 0.526 | 0.278 | 0.035 | 0.542 | 0.550 | 0.671 |
| **FFM (kg)** | β standardized | −0.252 | −0.191 | −0.209 | −0.006 | 0.320 | −0.459 | −0.209 | 0.016 | 0.001 | 0.186 | 0.147 | 0.053 | −0.204 | 0.068 | −0.220 | 0.313 |
| | *p* value | 0.171 | 0.304 | 0.259 | 0.974 | 0.128 | 0.024 | 0.259 | 0.940 | 0.515 | 0.232 | 0.347 | 0.736 | 0.212 | 0.679 | 0.178 | 0.052 |
| **BF%** | β standardized | 0.182 | −0.191 | 0.106 | −0.203 | −0.529 | 0.321 | 0.002 | −0.341 | −0.147 | 0.186 | −0.099 | −0.169 | 0.330 | −0.136 | 0.187 | −0.172 |
| | *p* value | 0.327 | 0.304 | 0.571 | 0.904 | 0.008 | 0.126 | 0.991 | 0.103 | 0.347 | 0.232 | 0.526 | 0.278 | 0.040 | 0.408 | 0.262 | 0.294 |
| **Waist Cir (cm)** | β standardized | 0.058 | −0.018 | 0.063 | −0.083 | 0.299 | 0.100 | −0.089 | −0.028 | 0.038 | −0.107 | 0.210 | −0.167 | 0.041 | 0.211 | −0.236 | 0.098 |
| | *p* value | 0.780 | 0.930 | 0.761 | 0.688 | 0.166 | 0.649 | 0.687 | 0.901 | 0.832 | 0.533 | 0.242 | 0.353 | 0.798 | 0.184 | 0.138 | 0.543 |
| **Hip Cir (cm)** | β standardized | −0.436 | 0.245 | 0.106 | −0.093 | 0.113 | −0.239 | 0.258 | −0.132 | 0.074 | 0.303 | −0.099 | −0.093 | −0.077 | −0.226 | 0.184 | 0.138 |
| | *p* value | 0.048 | 0.284 | 0.571 | 0.688 | 0.617 | 0.285 | 0.245 | 0.557 | 0.670 | 0.072 | 0.526 | 0.590 | 0.643 | 0.166 | 0.262 | 0.401 |

BMI: body mass index; FM: fat mass; FFM; fat free mass; BF%: body fat mass percentage; Cir: circumference; CH: carbohydrates; Prot: protein. Statistical significance, established at <0.05.

## 4. Discussion

The main findings of this study are that: (1) the MCT led to a reduction in adiposity and a maintenance of fat-free mass after 6 months in frail and pre-frail older people while no significant changes were observed in body composition in the CON; (2) 4 months of detraining after the MCT led to increases in adiposity and decreases in fat-free mass compared to values observed after MCT; and (3) energy intake showed different associations with changes in FM and BF% in CON and TRAIN, while carbohydrate distribution had a negative relationship with changes in BMI and FFM during detraining in CON.

The effects of multicomponent exercise programs on physical function of pre-frail and frail people have been analyzed in depth through a specific systematic review [27]; however, possible improvements in body composition have been less studied, especially in this specific population. In our study, a 6-month MCT reduced BMI, FM and BF%. In line with these findings, other authors also found significant changes [28–30], while results of Sousa et al. did not observe any changes in these variables [31]. Even in our study, frail and pre-frail older adults did not improve significantly their muscle mass after the 6-month intervention period, they may experience some benefits in muscle strength and function (unpublished data), probably because MCT causes neuromuscular adaptations improving muscle quality but not muscle mass [32], at least at this intensity and program duration. Studies mentioned above found improvements in muscle mass after this type of intervention, but they used dual-energy x-ray absorptiometry or computed tomography devices that could be more sensitive than BIA in detecting muscle changes after short training periods. Specifically, changes observed with these techniques were in appendicular lean mass [28], cross sectional muscle area and muscle fat-infiltration [30,33]. Moreover, our MCT was not specifically designed to improve body composition as its main objective was to improve functional capacity. In this regard, the use of repetition maximum (RM) as the optimal method to improve muscle mass in frail participants has been established [34]. However, we considered that to measure RM for all specific muscles involved in each exercise would not be useful for the replication in a real setting.

Some other reasons are proposed to explain why we found changes within groups, although our MCT did not improve body composition in TRAIN when compared to CON. One reason for this similar trend in both groups was our health-related talks. They were specially performed to improve adherence in CON and could be useful in raising awareness about their lifestyles. Moreover, the fact of being evaluated approximately every 3 months could be also have influenced them. This maintenance in CON and TRAIN groups reveals the importance of developing this type of intervention in pre-frail and frail older populations in order to prevent their expected decline in body composition. Therefore, developing strategies of this nature could help improve health in older people reducing the risk of adverse events and hospitalizations. In addition, the fact of including pre-frail older adults, as in our study, could mask significant results, since frail subjects are supposed to have less muscle mass and more adiposity and, therefore, they could have greater improvements in both parameters. Moreover, other aspects such as the large differences in methodologies, sample sizes and intervention durations among studies may explain the differences found between studies, which make it difficult to draw clear conclusions about the efficiency of this type of intervention for improving body composition in this population.

Although the benefits of MCT in health-related parameters of older people have been described in recent years [10,35], it is still unknown what happens when they leave the exercise programs. To the best of our knowledge, there is no specific study about body composition parameters in detraining, so, to date, this study is pioneering. According to our results, four months are enough to induce increases in FM, BF% and waist circumference and decreases in FFM in TRAIN once the intervention is stopped. As it is well known, an excess of adiposity contributes to an array of negative health outcomes, leading to significant morbidity, disability, decreased quality of life and, in some cases, increased rates of mortality in this population [36]. Due to the latter, these findings are of great relevance.

Besides, TRAIN also showed significant decreases in fat free mass, while this change was not observed in CON. Although our MCT did not significantly increase fat-free mass after 6 months, there was a small trend of increase, which may be more appreciable with a larger intervention period or more specific strength training. Considering these results, it seems that when the exercise stimulus for muscle stops, the decreases happens faster. As preserving muscle mass during aging has been a challenge for improving health since the development of sarcopenia is highly related with poor physical function, dependence and mortality [37,38]; smaller breaks in these types of interventions should be encouraged. Moreover, when a training program is stopped, this detriment not only affects body composition, but also other values return to baseline levels such as physical function [39] and metabolic parameters [40], as other authors have stated. Adding this knowledge to the literature is necessary given that the elder population frequently stops their activities as a consequence of illness and because exercise programs often stopped for 3 months following school calendars. Moreover, the current world situation due to the global pandemic caused by COVID-19 has led to a sudden stop in exercise programs; therefore, these findings are of great interest. In order to avoid reversibility, public institutions should give more opportunities and promote regular and continued physical activity practice and specific exercise programs for elders. In addition to this, further research regarding the optimal retraining time for regaining losses might be an interesting line to improve knowledge in this area.

It should be highlighted that aging-related physiological changes in body composition are complex and are not influenced only by exercise. Factors such as time spent in sedentary activities, diet or pharmacological intake may play an important role in these changes. In this study, we aimed to examine if nutritional intake is related to the changes observed after MCT and detraining periods, as it has been related to body composition changes and physiopathologies previously named [13,41]. Parameters included in this study were total energy intake and macronutrient distribution because of their contribution in energy imbalance, their consequent effects on body composition and the relation with frailty [42]. In this way, we found that energy intake is related to an increase in FM and BF% in TRAIN during detraining. Probably it could be due to a positive energy balance which could had been compensated with a higher energy expenditure during the exercise period. Opposite, in CON this energy balance seems to have a negative association; however, other variables not considered in the analyses such as daily physical activity could be influencing the results. Quality of diet or even different distribution in macronutrients intake could also be affecting this relationship. For instance, a small consumption of carbohydrates may lead to an increase in other macronutrients such as protein intake which may be increasing FFM and consequently BMI.

Nevertheless, no associations were found with protein and other nutritional parameters with changes in body composition variables in our frail and pre-frail seniors as other authors have previously shown in a large population [43]. Thus, other dietary aspects such as food quality or dietary patters could be influential in some of these changes and should be studied in detail. In addition, it should be studied if the effects of MCT on body composition could be improved or if detrimental changes during detraining periods could be slowed down by dietary control, as new investigations are being performed combining exercise and dietary intervention or even supplementation [44]. In fact, effects of multidisciplinary strategies including nutritional education have been demonstrated that could be effective for improving not only body composition-related diseases but also costs [45].

Some limitations of this study should be mentioned. Firstly, a randomization of the sample was not possible as it was difficult to change older adult routines and some participants refused to participate in TRAIN group. Therefore, the sample was divided into CON and TRAIN groups according to the volunteers' preferences/availability in order to maximize training attendance, which is needed to elucidate the effectiveness of this MCT protocol. Analysis was not conducted under the intention to treat framework which may lead to time bias or survival bias. Even though our study has one of the largest sample sizes related to the effects on MCT in body composition of frail and pre-frail older people, it is still reduced, which could lead to a lack of statistical power, especially in dietary

intake-related analyses. Therefore, studies with larger number of participants should be developed to establish deeper conclusions. Body composition measurements in this study were made using BIA and although we have tried to standardize all measurements to avoid possible bias, this method has been criticized and its accuracy is under discussion [46]. Regarding diet, institutionalized participants follows a standardized diet, which could be hiding different results. It should be highlighted that to the best of our knowledge, our MCT is the one with the longest intervention in this topic. Even though no one in TRAIN revoked their participation and the percentage of attendance was high, it was difficult to encourage attendance for CON, which may lead to bias. Moreover, all participants (both in CON and TRAIN groups) have requested to participate again in a new phase of the project. Thus, MCT may be an effective strategy to increase exercise adherence in this population.

## 5. Conclusions

Our 6-month MCT seems to reduce adiposity and maintain muscle mass of frail and pre-frail older people. A detraining period of 4 months leads to an increase in adiposity and a decrease in muscle mass. Moreover, total energy intake seems to have a negative association on FM and BF% changes in TRAIN, opposite to CON. A higher percentage of carbohydrate seems to have a negative contribution in CON in FFM and BMI during the detraining. The MCT should continue as long as possible to keep the positive changes because after detraining, some of the positive changes reversed. The fact that similar results were found in CON and TRAIN groups, highlights the importance for implementing and promoting interventions including MCT in frail and pre-frail elders, as both seem to be effective to preserve body composition. Future research including nutritional interventions or focusing on the maximum period of detraining without detrimental effects on body composition could help to prevent pathologies, dependence, or the need of care and improve not only quality of life in this population but also individual and social sustainability.

**Author Contributions:** Conceptualization, A.M., J.A.C., A.G.-C. and G.V.-R.; data curation, A.M. and J.M.-P.; formal analysis, A.M., J.M.-P. and A.G.-C.; funding acquisition, I.A., J.A.C. and G.V.-R.; investigation, A.M., D.N.-V., Á.I.F.-G., L.S.-R., J.M.-P. and A.G.-C.; methodology, A.M., J.M.-P., J.A.C., A.G.-C. and G.V.-R.; project administration, J.A.C., A.G.-C. and G.V.-R.; resources, A.M., D.N.-V., Á.I.F.-G., L.S.-R. and A.G.-C.; supervision, J.A.C., A.G.-C. and G.V.-R.; validation, J.P.-G., E.G. and I.A.; visualization, A.M., L.S.-R., J.P.-G., E.G. and I.A.; writing—original draft, A.M. and A.G.-C.; writing—review and editing, D.N.-V., A.I.F.-G., L.S.-R., J.M.-P., J.P.-G., E.G., I.A., J.A.C. and G.V.-R. All authors have read and agreed to the published version of the manuscript.

**Funding:** This study was funded by "Centro Universitario de la Defensa de Zaragoza" (UZCUD2017-BIO-01) and "Ministerio de Economía, Industria y Competitividad" (DEP2016-78309-R). Biomedical Research Networking Center on Frailty and Healthy Aging (CIBERFES) and FEDER funds from the European Union (CB16/10/00477).

**Acknowledgments:** The authors are grateful to all the collaborators, nursing homes, health centers and participants whose cooperation and dedication made this study possible. A.M. received a PhD grant from "Gobierno de Aragón" (2016–2021). D.N-V. received a grant from "Gobierno de Aragón" (DGAIIU/1/20). Á.I.F.-G received a grant from the Spanish Government (BES-2017-081402).

**Conflicts of Interest:** The authors declare no conflict of interest.

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
