# Peer review of "Effects of a Multicomponent Exercise Program, a Detraining Period and Dietary Intake Prediction of Body Composition of Frail and Pre-Frail Older Adults from the EXERNET Elder 3.0 Study"

_sustainability, doi:10.3390/su12239894_

Round 1

Reviewer 1 Report

This study tries to evaluate the impact of a multicomponent training (MCT) program and its follow-up on body composition of frail and pre-frail older adults. In addition, they aimed to assess the association of these changes with energy and macronutrient intake. Although the benefits of MCT on physical and mental health have been clearly stablished, its impact on body composition have been frequently overlooked.  Taking into account, the current state of the art, the study seems relevant. However, I consider that the manuscript need some changes before being published.

Abstract

Line 31. Authors stated that they aimed “to assess the influence of energy and macronutrient intake in these changes”.  In my opinion, the design of the study does not allow answering this question. The authors should tone down this aim.

Introduction

Lines 50-66. I believe that it is more appropriate to explain first the effects of ageing and then frailty.

Lines 69-70. The statement “different type of training may lead to …… and not all population may respond in the same way” should include a reference.

Lines 90-91. “To examine if nutritional intake acts a mediator in the changes….. “ needs a mediation analysis. I consider most adjusted to the methodology “to examine if nutritional intake is related to the changes observed after the ….

Methods

Lines 109. The statement “In this report, we have included the last TWO evaluation” is confusing. Throughout the article, three evaluations are presented.

Line 181. When was the dietary intake assessed? At the beginning of the intervention?

Line 207. It is not clear, how the linear regressions were calculated. Did they perform four different simple linear regression (one for each variable) or only one multiple linear regression with 4 variables? Taking into account the results, I assume that they selected the second option. However, authors should clarify this topic.

Results.

Line 229. Is the term “detraining” appropriate for the control group?

Line 255. Total energy and macronutrient intake did not INFLUENCE the variation…… In my opinion, current design does not seem to be able to ascertain “influence”.

Table 2. The number of participants assessed in each group and each time point should be stated. Were represented results of those participants who were assessed in post-detraining? Or those who finished the training period were analysed for the effects of training?

Table 3. Some standardised beta value are higher than 1. It often happens when there is a high degree of multicollinearity, as it might be the case of the percentage of macronutrients; if one eats more fat and CF the intake of proteins decrease. Perhaps, 4 simple lineal regressions with each macronutrient or total energy intake as independent variables could be more appropriate.

Table 3.  The authors should specify the units or percentages for CH, Fat or proteins

Table 3. The number of participants for each association should be stated.

Discussion.

Lines 265-269. The authors stated that MCT leaded to maintenance of adiposity and detraining increased adiposity. Considering these results fat mass should be higher after detraining than before training. However, they are similar (28.8 vs 28.9). The authors should consider mentioning the trend of decreasing fat mass after MCT.

Limitations of the study. The fact that the groups were not randomized is a clear limitation of the study. In addition, the fact that people in the control group revoked participation more than in the intervention group could also biased the results. Authors recognized that sample size is a limitation for the effects of MCT on body composition. However, sample size is specially limiting for assessing the association between dietary intake and changes in body composition, mainly in the control group (only 19 participants for 4 variables).

Reviewer 2 Report

This study aims to evaluate an interesting topic, the impact of detraining after exercise program for frail older adults. However, it is questionable if the study design and statistical methods can answer that important question. First, the two groups are not randomly assigned. Without randomized groups, it carries significant selection bias, which can't conclude a true causality of the intervention. Second, the analysis was not conducted under the intention to treat frame. They only included those who finished the follow up for the final analysis, which will also lead serious confounding, lead time bias or survival bias. Third, it is questionable if ANOVA would be an appropriate analysis. To compare the two groups for multiple times during follow up, mixed effect model would have been more appropriate, but it also has to be clearly mentioned which comparison they wanted to investigate. After reading the objectives and final results, it was not clear what kind of comparison they wanted to do. Was it to compare the two groups? or was it to compare the two time frame within the group? Lastly, overall grammar and style of writing have to be improved for better understanding. Abstract is hard to understand without full spelling of the abbreviation, or too many abbreviation. 

Reviewer 3 Report

Dear authors

Some corrections should be introduced.

Round 2

Reviewer 1 Report

The authors have improved the manuscript considerably. However, I think that the article needs further changes, mainly to clarify the results after their statistical analysis.

Title: The authors deleted the term "mediation" from the aim. Accordingly, this term should be also changed in the title.

Statistical analysis: In simple regression analysis is not relevant to specify that data were introduced by "enter" method, because there is only one variable.

Results

Table 2: It seems contradictory that only 39 people were measured at the end of the detraining in the TRAIN group. However, 43 people completed a total of 10 months. In the control group, 19 people ended the study and only 17 people were analyzed for the whole period.  Additionally, the values in 10 months and the sum of 6 and 4 months of many variables are different. For example,  in WC for the TRAIN group, the sum is 6.96, and the value for 10 months 5.08.

Table 3. There are some discordances between beta values and p. In the control group, the association between changes in Hip Cir and total energy during training seems significant. Beta value for the association between total energy and changes in BMI during detraining in the TRAIN group is too small to be significant.

Discussion

First paragraph. The sentence "Body composition tend to worsen in the control group" is not accurate because there are small decreases in BMI, FM and BF

Throughout, the discussion there is an apparent contradiction between the idea that MCT reduced adiposity and control tended to increase it and the statement that MCT did not improve body composition in TRAIN when compared to CONTROL. This should be due to different statistical comparisons. I believe that the authors should clarify this issue because the message is not clear.

Fat-free mass or muscle mass should be consistently used in the whole manuscript. 

Conclusions:

Carbohydrate distribution should be change by a higher percentage of carbohydrates in the diet seems to have.

Similar results in control and TRAIN groups? I believe that there are substantial differences as the authors stated in the discussion.

Reviewer 2 Report

Thank you for the opportunity to review this revised article. The authors reflected the reviewer's questions and concerns appropriately and the manuscript significantly improved. The methodology and the objectives are now more clearly stated. 

Here are minor suggestions:

Abstract needs significant editing. MCT is not spelled in the abstract. There are many "changes" in description of the results, but the direction is not clearly stated. Imaging that readers will only read the abstract, and they would like to know the direction of the changes. Therefore, abstract itself should be able to speak by itself about he results. Similarly, it is not clearly stated if the training duration was 6 months or not, and if the detraining was 4 months or not in the abstract. 

Throughout the manuscript, avoid using "elderly" since this is not recommended term anymore in geriatrics society. Instead, use "older adults".

I think one of the key message is that it is important to continue MCT as long as possible to keep the positive changes. AFter detraining, some of the positive changes reversed. It is stated in the abstract, but I think that it also has to be clearly mentioned in conclusion section of the manuscript. 
